# Voltage imaging reveals the dynamic electrical signatures of human breast cancer cells

Peter Quicke[1,2,8], Yilin Sun[1,8], Mar Arias-Garcia [3], Melina Beykou[3,4], Corey D. Acker[5], Mustafa B. A. Djamgoz[6,7], Chris Bakal [3✉] & Amanda J. Foust [1✉]

Cancer cells feature a resting membrane potential ($V_m$) that is depolarized compared to normal cells, and express active ionic conductances, which factor directly in their pathophysiological behavior. Despite similarities to 'excitable' tissues, relatively little is known about cancer cell $V_m$ dynamics. Here high-throughput, cellular-resolution $V_m$ imaging reveals that $V_m$ fluctuates dynamically in several breast cancer cell lines compared to non-cancerous MCF-10A cells. We characterize $V_m$ fluctuations of hundreds of human triple-negative breast cancer MDA-MB-231 cells. By quantifying their Dynamic Electrical Signatures (DESs) through an unsupervised machine-learning protocol, we identify four classes ranging from "noisy" to "blinking/waving". The $V_m$ of MDA-MB-231 cells exhibits spontaneous, transient hyperpolarizations inhibited by the voltage-gated sodium channel blocker tetrodotoxin, and by calcium-activated potassium channel inhibitors apamin and iberiotoxin. The $V_m$ of MCF-10A cells is comparatively static, but fluctuations increase following treatment with transforming growth factor-$\beta$1, a canonical inducer of the epithelial-to-mesenchymal transition. These data suggest that the ability to generate $V_m$ fluctuations may be a property of hybrid epithelial-mesenchymal cells or those originated from luminal progenitors.

[1] Department of Bioengineering, Imperial College London, London SW7 2AL, UK. [2] The Francis Crick Institute, London NW1 1AT, UK. [3] Institute of Cancer Research, Cancer Biology, London SW3 6JB, UK. [4] Department of Electrical and Electronic Engineering, Imperial College London, London SW7 2AZ, UK. [5] University of Connecticut School of Medicine, R. D. Berlin Center for Cell Analysis and Modeling, Farmington, CT, USA. [6] Department of Life Sciences, Imperial College London, London SW7 2AZ, UK. [7] Biotechnology Research Centre, Cyprus International University, Haspolat, Nicosia, TRNC Mersin 10, Turkey. [8] These authors contributed equally: Peter Quicke, Yilin Sun. ✉email: chris.bakal@icr.ac.uk; a.foust@imperial.ac.uk

All cells in the body exhibit a voltage difference ($V_m$) across the plasma membrane which regulates a wide range of functions such as gene expression, secretion, and whole-cell motility. Cellular $V_m$ at rest varies both between and within cell types. Interestingly, whilst this is ca. −70 mV in mature 'quiescent' cells, including nerves and muscles, it is noticeably depolarized ($V_m$ ca. −50 to −10 mV) in proliferating cells, including cancer cells and stem cells[1,2].

$V_m$ fluctuates dramatically, both spontaneously and in response to stimuli, in classically excitable tissues such as the heart, muscle, and nerve, which support the generation and conduction of action potentials. The resting $V_m$ of several cell types has been shown to fluctuate[3]. These include cells with rhythmic activity, e.g. neurons controlling respiration[4], arterial vasomotion[5], biological 'clocks'[6], and sleep[7,8]. Oscillations of $V_m$ also manifest in pathophysiological situations, such as epilepsy and neuronal degeneration, and can extend to network effects[9,10].

In several carcinomas, functional expression of voltage-gated sodium channels (VGSCs) promotes the metastatic process[11]. Treating carcinoma cells in-vitro with VGSC blockers partially suppresses 3D invasion[12,13]. The most specific inhibitor of VGSCs is tetrodotoxin (TTX), which blocks the channel by binding to a site within the channel pore when the channel is in the open state[14]. TTX reduces invasion in carcinoma cells in-vitro, and this effect is abolished by siRNA silencing of the VGSC Nav1.5 in-vivo[12,13,15,16]. Gradek et al. recently demonstrated that silencing of SIK1 induces Nav1.5 expression, invasion, and the expression of the epithelial-to-mesenchymal transition (EMT)-associated transcription factor SNAI1[17]. However, as noted above, the steady-state resting $V_m$ of human breast cancer cells relative to normal epithelia is strongly depolarized[18]. In the case of the MDA-MB-231 cells, derived from a highly aggressive triple-negative breast cancer (TNBC), $V_m$ rests between −40 and −20 mV[15,19,20]. The $V_m$-dependent inactivation of VGSCs means that the majority of channels should be permanently inactivated at such depolarized membrane potentials and therefore insensitive to TTX. Nevertheless, TTX has been shown repeatedly to inhibit the invasiveness of these cells and several other carcinomas[11,15,19,21–24], potentially by blocking the persistent window current[19,25].

Although the depolarization of resting $V_m$ and the enriched VGSC expression in aggressive cancer cell lines are established (reviewed by Yang and Brackenbury[2] and Pardo and Stühmer[26]), unlike classical excitable tissues (e.g., heart, muscle, nerve), comparatively little is known about cancer's $V_m$ dynamics. Studies utilizing multi-electrode arrays detected $V_m$ fluctuations but could not attribute them to individual cells[27,28]. Here, in contrast, we captured cellular-resolution, spatially resolved $V_m$ dynamics in human breast cancer cells with a fast, electrochromic voltage-sensitive dye, enabling optical monitoring of $V_m$ changes in hundreds of cells simultaneously. Through an unsupervised machine learning protocol, we classified and characterized the dynamic electrical signatures (DESs) of the cellular $V_m$ time series obtained with high-throughput imaging. A subset of MDA-MB-231 breast cancer cells exhibited hyperpolarizing "blinks" and "waves", in contrast with the quiescent, static $V_m$ of non-tumorigenic MCF-10A cells. Application of TTX suppressed the $V_m$ fluctuations in MDA-MB-231 cells whilst treatment of MCF-10A cells with transforming growth factor-β1 (TGF-β), which stimulates EMT, induced $V_m$ fluctuations in these cells. Taken together, these data suggest that the ability to generate $V_m$ fluctuations is acquired during the EMT and may participate in cancer progression.

## Results

**Di-4-AN(F)EP(F)PTEA fluorescence ratio linearly reports change in $V_m$.** We imaged the membrane potential of cultured cell monolayers with extracellularly applied di-4-AN(F)EP(F)PTEA, a dye that inserts into the outer membrane, shifting its absorption and emission spectra as a function of membrane potential with sub-microsecond temporal fidelity[29]. We sequentially excited the dye with blue and green light-emitting diodes (LEDs, Figs. 1a, b and S1), taking the ratio of fluorescence excited by each color at each point in time and dividing by the baseline ratio ($\Delta R/R_0$). A change in $V_m$ causes the fluorescence excited by each color to change in opposite directions (Fig. 1c), amplifying the corresponding change in the ratio. The ratiometric imaging scheme also partially mitigates the confounds of uneven dye labeling, photobleaching decay, and mechanical motion[30]. This approach enabled us to image the dynamics of hundreds of human breast cancer cells simultaneously with cellular resolution.

We first verified that the fluorescence ratio ($\Delta R/R_0$) linearly reported changes in $V_m$. By imaging Di-4-AN(F)EP(F)PTEA fluorescence, while stepping $V_m$ through a range of values in whole-cell voltage clamp of MDA-MB-231 cells, we observed that the ratio of blue-to-green Di-4-AN(F)EP(F)PTEA fluorescence varied linearly with changes in $V_m$ over a physiological range of $V_m$ from −60 to +30 mV. The slope of $\Delta R/R_0$ vs. $V_m$ showed an average sensitivity of 5.1 ± 0.43% per 100 mV (mean ± standard error of the mean (s.e.m.); $n = 12$ cells; Fig. 1d–g). Furthermore, as expected, global depolarization of all cells on the coverslip by washing in a high-potassium (100 mM) extracellular solution also increased $\Delta R/R_0$ (Fig. S2). These results show that the ratio $\Delta R/R_0$ faithfully reports $V_m$ changes in subsequent recordings of spontaneous $V_m$ fluctuations (Fig. 1h).

**Membrane voltage fluctuates in human breast cancer cells.** We compared the frequency of optically detected, transient, negative-going events ("−VEs") in non-tumorigenic MCF-10A cells to that of eight human breast cancer cell lines: MDA-MB-231, MDA-MB-468, Cal-51, SUM-159, Hs578T, MDA-MB-453 and BT-474, and T-47D. Figure 2 displays the rate of −VEs for each field of view imaged. All of the cancer cell lines exhibit a −VE rate significantly greater than that of the benign MCF-10A breast epithelial line: MDA-MB-231 ($p = 1.4 \times 10^{-8}$), MDA-MB-468 ($p = 1.1 \times 10^{-8}$), Cal-51 ($p = 1.1 \times 10^{-5}$), SUM-159 ($p = 3.4 \times 10^{-3}$), Hs578T ($p = 5.7 \times 10^{-6}$), MDA-MB-453 ($p = 5.2 \times 10^{-5}$), BT-474 ($p = 2.2 \times 10^{-11}$), and T-47D ($p = 0.01$). The $p$-values are for the comparison of the average per field-of-view negative event "−VE" rate for each cancer cell line compared to the non-tumorigenic MCF-10A line using a Mann–Whitney $U$ test with Benjamini–Hochberg correction for multiple comparisons (false discovery rate = 0.05). Strikingly, the extent of activity correlated strongly with the line sub-type. Cancer lines previously classified as Luminal B (BT-474, MDA-MB-453) or Basal A (MDA-MB-468) were the most active[31] (Fig. 2). But while Luminal A and Basal B lines were less active than Luminal B or Basal A, they were nonetheless markedly more active than non-transformed cells.

The following sections optically characterize $V_m$ dynamics of the MDA-MB-231, a widely studied model of metastatic, triple-negative breast cancer.

**Cellular-resolution membrane voltage fluctuations in MDA-MB-231 cells.** A subset of MDA-MB-231 cells exhibited fluctuating $V_m$. We imaged spontaneous $V_m$ fluctuations at 5 frames/s in cultured monolayers of the highly aggressive triple-negative breast cancer cell line MDA-MB-231 (Fig. 3a). 6.84 ± 0.97% ($n = 22$ coverslips) of cells on each coverslip displayed a fluctuating $V_m$ (Fig. 3b, c). The great majority (>90%, 91/100) of high signal-to-noise ratio, large amplitude (>|1.5|$\Delta R/R_0$) events were negative-going ("−VE", Fig. 3b). Therefore, subsequent analyses focus on the −VEs (Fig. 2a, b). Most cells exhibited few or no

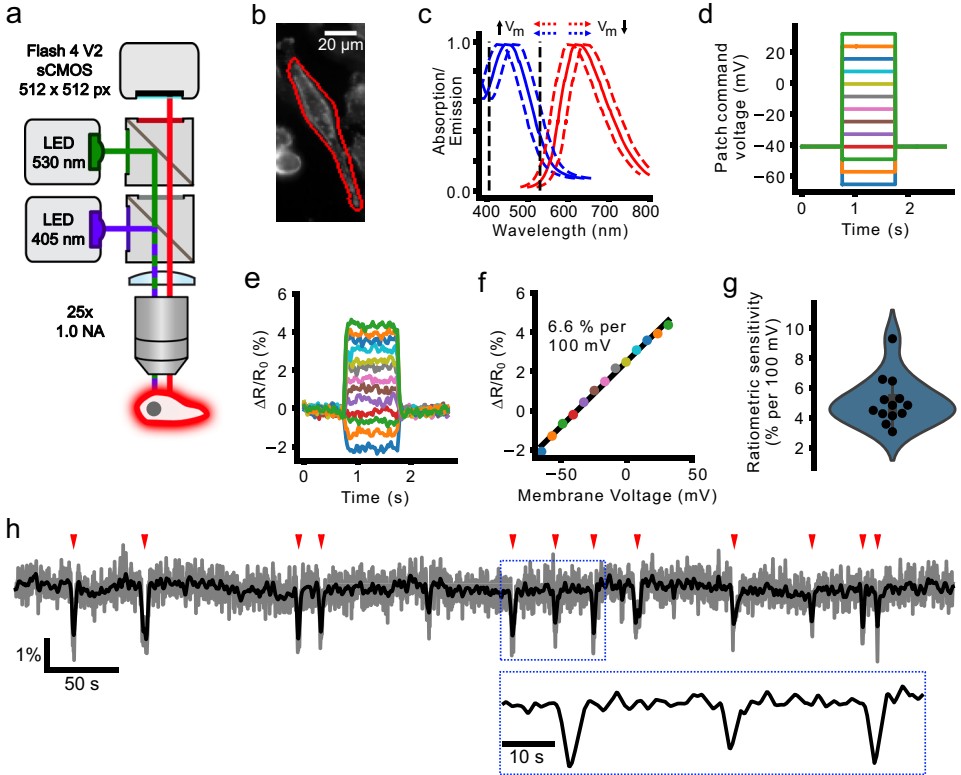

**Fig. 1 Electrochromic imaging of MDA-MB-231 cells with voltage dye di-4-AN(F)EP(F)PTEA. a** Schematic of the widefield epifluorescence imaging system with two-color excitation. **b** MDA-MB-231 cell stained with di-4-AN(F)EP(F)PTEA with simultaneous voltage clamp **c** The blue and red lines show di-4-AN(F)EP(F)PTEA's excitation and emission spectra[29], respectively. Fluorescence is excited by blue and green LEDs on opposite sides of the excitation maximum (black vertical dashed lines). Decreasing or increasing $V_m$ causes the excitation spectra to shift right or left, respectively. Decreasing $V_m$, therefore, causes reduced emission in response to blue channel excitation, and increased fluorescence with green channel excitation and vice versa for increasing $V_m$. **d** Waveforms commanding the MDA-MB-231 whole-cell voltage clamp for calibration of the ratio of blue- to green-excited fluorescence with respect to the baseline ($\Delta R/R_0$). **e** Recorded $\Delta R/R_0$ signal in response to the injected $V_m$ waveforms. **f** Average $\Delta R/R_0$ change with membrane voltage change and a linear fit to the data indicating the imaging sensitivity (% change in ratio per 100 mV membrane voltage change). **g** Measured sensitivities for different patched cells with a gaussian kernel estimate (blue envelope). **h** Example time course of a spontaneously active MDA-MB-231 cell displaying typically observed transient $V_m$ hyperpolarisations (indicated by red ticks). Gray, unfiltered time course, black, gaussian filtered time course, sigma = 3 sampling points (0.6 s).

fluctuations, but a subset of cells was highly active (Fig. 3c, see also Supplemental Movie 1). Among the active cells, events were detected at an average frequency of $2 \pm 0.2$ events/cell/1000 s ($n = 20$ coverslips, Fig. 3d).

We then sought to describe the "dynamic electrical signature" (DES) of individual cells based on their $V_m$ imaging time series. A DES is a multi-parametric feature vector that captures various aspects of $V_m$ fluctuations over time such as power spectral properties, successive differences, and entropy. A DES captures variations beyond the event-based metrics and can be used to classify patterns based on their similarity. To describe the DESs, we implemented an unsupervised machine learning pipeline (Fig. 4). The Cellular DES Pipeline uses the Catch-22 algorithm[32] to extract features from individual $V_m$ traces. Feature extraction and hierarchical clustering on the "active" cellular time series, those in which $V_m$ fluctuations were detected (287 out of 2993, Fig. 4a–c), yielded silhouette coefficients indicating that the time series clustered into 3 or 4 classes (Fig. 4d). Manual inspection of the time series revealed higher inner-cluster similarly with 4 clusters. We named the DES classes identified by the 4 cluster classification: small blinking (blinking-S), waving, noisy, and large blinking (blinking-L). Figure 5 displays exemplar time series from each class. Cells of the four classes were observed simultaneously in the imaged fields-of-view (FOVs). Together these four DES

classes describe the temporal heterogeneity of $V_m$ fluctuations in MDA-MB-231 cells.

**TTX decreases $V_m$ fluctuations in MDA-MB-231 cells**. To assess the role of VGSC activity in the DES of MDA-MB-231 cells, we treated cultures with tetrodotoxin (TTX), a potent and specific inhibitor of VGSCs[14]. The ability of electrochromic dyes to report the effects of TTX is well established[33]. TTX decreased the frequency of $V_m$ fluctuations, especially of large amplitude hyperpolarizations (−VEs, Fig. 6a, b), in a dose-dependent manner. In particular, 10 µM TTX decreased the mean event rate by ~4×, from $9.5 \times 10^{-5}$ to $1.97 \times 10^{-5}$ events/cell/s ($p < 10^{-6}$, one-sided bootstrapped significance test on the mean negative event rate per cell; pre-TTX $n = 1063$ cells, 4 slips; with TTX $n = 2028$ cells, 4 slips; Fig. 6c). 1 µM TTX decreased the mean event rate by a lesser factor of ~ 2×, from $1.04 \times 10^{-4}$ to $4.99 \times 10^{-5}$ events/cell/s ($p = 0.019$, one-sided bootstrapped significance test on the mean negative event rate per cell; pre-TTX $n = 519$ cells, 2 slips; with TTX $n = 722$ cells, 2 slips; Fig. 6d). The effect of TTX on event frequency recovered following washout (10 µM TTX; pre-TTX $n = 486$ cells, 3 slips; with TTX $n = 1066$ cells, 3 slips; washout $n = 1005$ cells, 3 slips; Fig. 6e). For the feature-based analysis, projection of active TTX-treated MDA-MB-231 cells onto the active untreated cells' principal

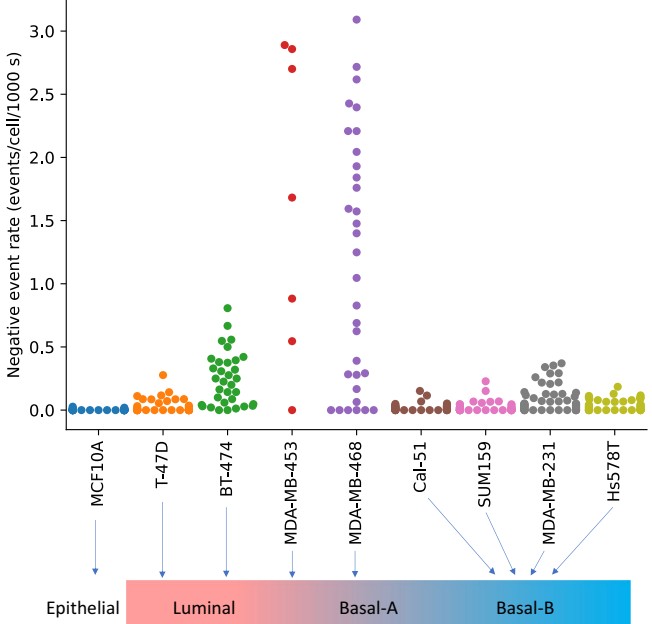

**Fig. 2 $V_m$ imaging reveals the dynamic membrane voltage of human breast cancer cell lines.** Each circle indicates the mean negative event "−VE" rate (per cell, per 1000 s) for all cells in the field-of-view of one imaging trial. All of the cancer cell lines exhibit a −VE rate significantly greater than that of the benign MCF-10A breast epithelial line ($p$-values all ≤0.01. The $p$-values are for the comparison of the average per field-of-view −VE rate for each cancer cell line compared to the non-cancerous MCF-10A line using a Mann–Whitney $U$ test with Benjamini–Hochberg correction for multiple comparisons (false discovery rate = 0.05).

component (PC) space did not reveal DES class differentiation with TTX (Fig. S3).

**Calcium-activated potassium channel inhibitors decrease $V_m$ fluctuations in MDA-MB-231 cells.** We assessed whether calcium-activated potassium (KCa) channels play a role in transducing calcium fluctuations into the optically measured $V_m$ fluctuations. We treated cultures with either 100 nM iberiotoxin (IbTx), a large-conductance KCa channel (BK) blocker, or 100 nM apamin, a small-conductance KCa channel (SK) blocker. Both IbTx and apamin decreased the frequency of −VEs by ~5× (Fig. 7). 100 nM IbTx decreased the mean −VE rate from $11.1 \times 10^{-5}$ to $1.94 \times 10^{-5}$ events/cell/s ($p < 10^{-6}$, one-sided bootstrapped significance test on the mean negative event rate per cell; pre $n = 2761$ cells, 6 slips; with IbTx $n = 1599$ cells, 6 slips). 100 nM apamin decreased the mean −VE rate from $10.9 \times 10^{-5}$ to $1.89 \times 10^{-5}$ events/cell/s ($p < 10^{-6}$, one-sided bootstrapped significance test on the mean −VE rate per cell; pre $n = 3110$, 6 slips; with apamin $n = 2223$, 5 slips). Separate imaging sessions of MDA-MB-231 cells in which IbTx and apamin were washed out following 45–60 min of application showed −VE rates comparable to the pre-IbTx and pre-apamin recordings. Following washout of 100 nM IbTx, the mean −VE rate was $18.6 \times 10^{-5}$ events/cell/s ($p = 0.997$, 2-sided bootstrapped significance test on the mean −VE rate compared to pre-apamin trials, washout $n = 817$ cells, 1 slip). Following washout of 100 nM apamin, the mean −VE rate was $15.8 \times 10^{-5}$ events/cell/s ($p = 0.934$, 2-sided bootstrapped significance test on the mean −VE rate compared to pre-apamin trials; washout $n = 1379$ cells, 1 slip). Thus, the effects of the KCa channel blockers were reversible.

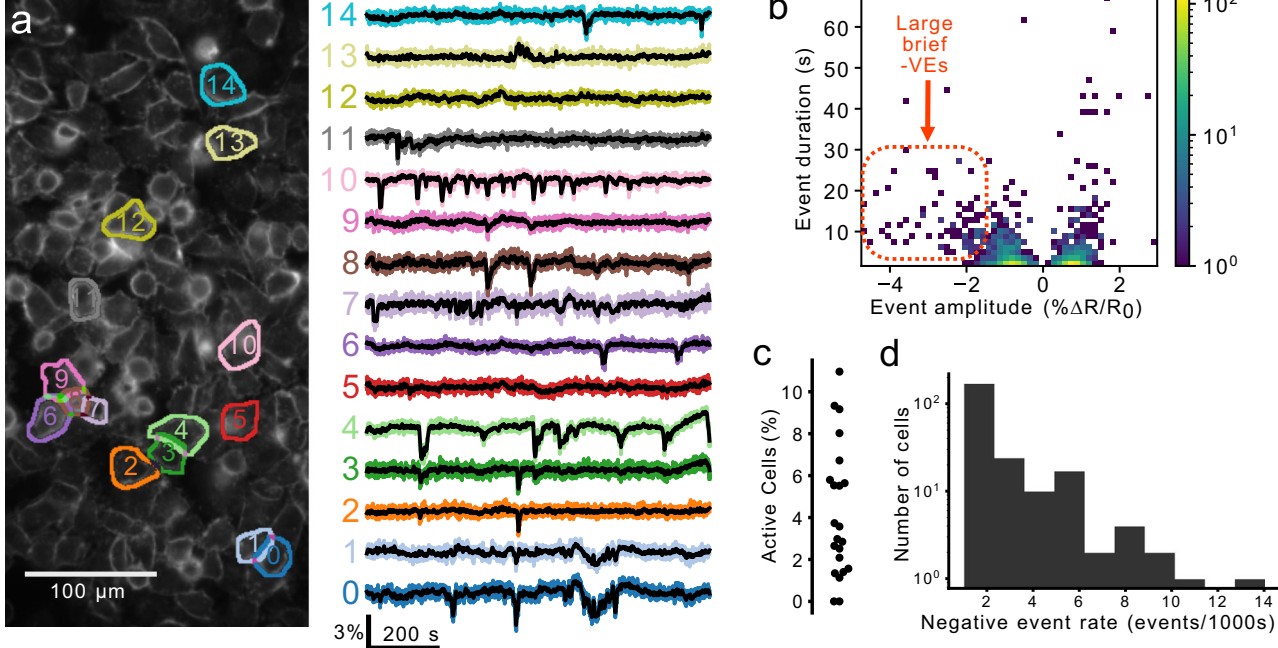

**Fig. 3 A subset of MDA-MB-231 cells exhibit $V_m$ fluctuations. a** The ratio of Di-4-AN(F)EP(F)PTEA fluorescence ($\Delta R/R_0$) was imaged at 5 frames/s in cultured MDA-MB-231 cells. The $\Delta R/R_0$ time series extracted from the segmented cells (right) reveals heterogeneous $V_m$ fluctuations consisting primarily of transient hyperpolarizations. **b** A log-scaled 2D histogram of event amplitude and duration. These fluctuations vary in their polarity (positive-going, "+VE" vs. negative-going "−VE"), amplitude, and duration. Large amplitude fluctuations were typically hyperpolarising (−VE amplitude) and short in duration. **c** Proportion of active cells for 20 technical replicates displaying the high variability in the proportion of active cells. **d** A log-scaled histogram of the mean −VE rate for all active cells (at least one −VE observed) showing the long-tailed distribution with a few highly active cells.

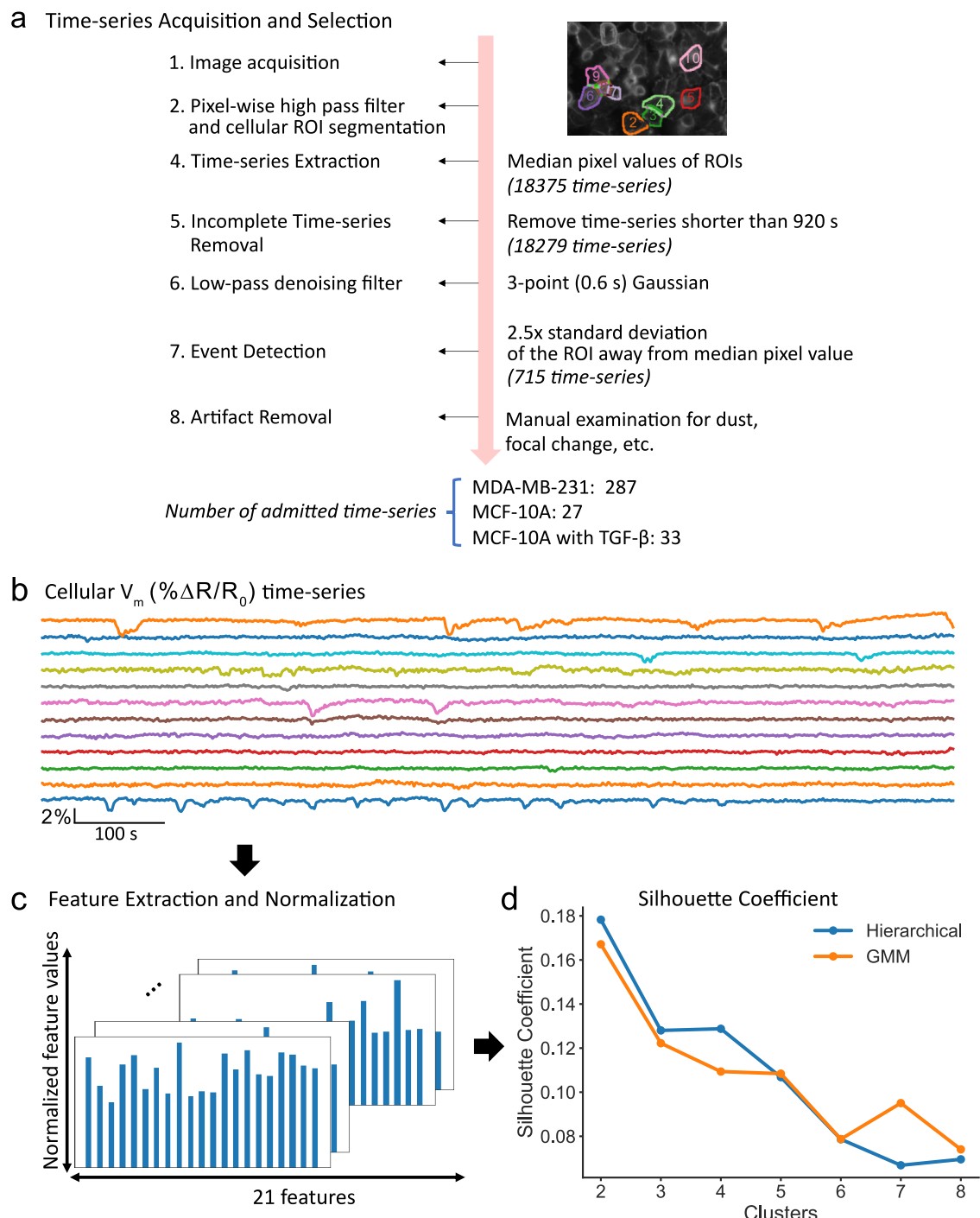

**Fig. 4 The dynamic electrical signature (DES) of MDA-MB-231 cells clustered into four classes. a** Time series were admitted to the unsupervised machine learning clustering pipeline if their temporally filtered $\Delta R/R_0$ exhibited fluctuations > 2.5× of their within-ROI pixel standard deviation from the median pixel value, and if also free of imaging artefacts including floating dust, mechanical vibration, and illumination edge effects. **b** Visual inspection of each cell's time series finalized admission to the cellular DES pipeline. **c** 22 features relevant to time series temporal patterns were extracted from each time series using the Catch-22 algorithm[32] and normalized with a Box-Cox transformation relative to their values. **d** The silhouette coefficients for different cluster numbers were generated through hierarchical clustering (blue) or gaussian mixture modeling (GMM, orange) on the normalized features.

**$V_m$ dynamics in MCF-10A cells**. To assess the impact of MDA-MB-231's cancerous, aggressive phenotype on $V_m$ dynamics, we compared it to the $V_m$ dynamics of non-tumorigenic MCF-10A cells (Fig. 8a, c). Applying the same imaging protocol, we observed that only a small subset ($0.46\% \pm 0.14\%$, $n = 6944$ cells, 12 coverslips) of MCF-10A cells exhibited $V_m$ fluctuations compared to MDA-MB-231 cells ($6.84\% \pm 0.97\%$, $n = 3017$ cells, 13 coverslips, Fig. 8e). Interestingly, incubation of the cells with

TGF-$\beta$1 (5 ng/mL), a growth factor known to stimulate EMT[34], increased the percentage of cells exhibiting $V_m$ fluctuations to $0.81 \pm 0.19\%$ ($n = 3787$ cells, 13 coverslips, Fig. 8b, d). With TGF-$\beta$, the mean $-$VE rate increased from $2.85 \times 10^{-6}$ to $1.4 \times 10^{-5}$ events/cell/s ($p = 0.000567$, one-sided bootstrap difference of means at the cell level, Fig. 8c–e).

Visualization of the MCF-10A cells' normalized features in 2D PC space shows the separation of DESs between TGF-$\beta$-treated

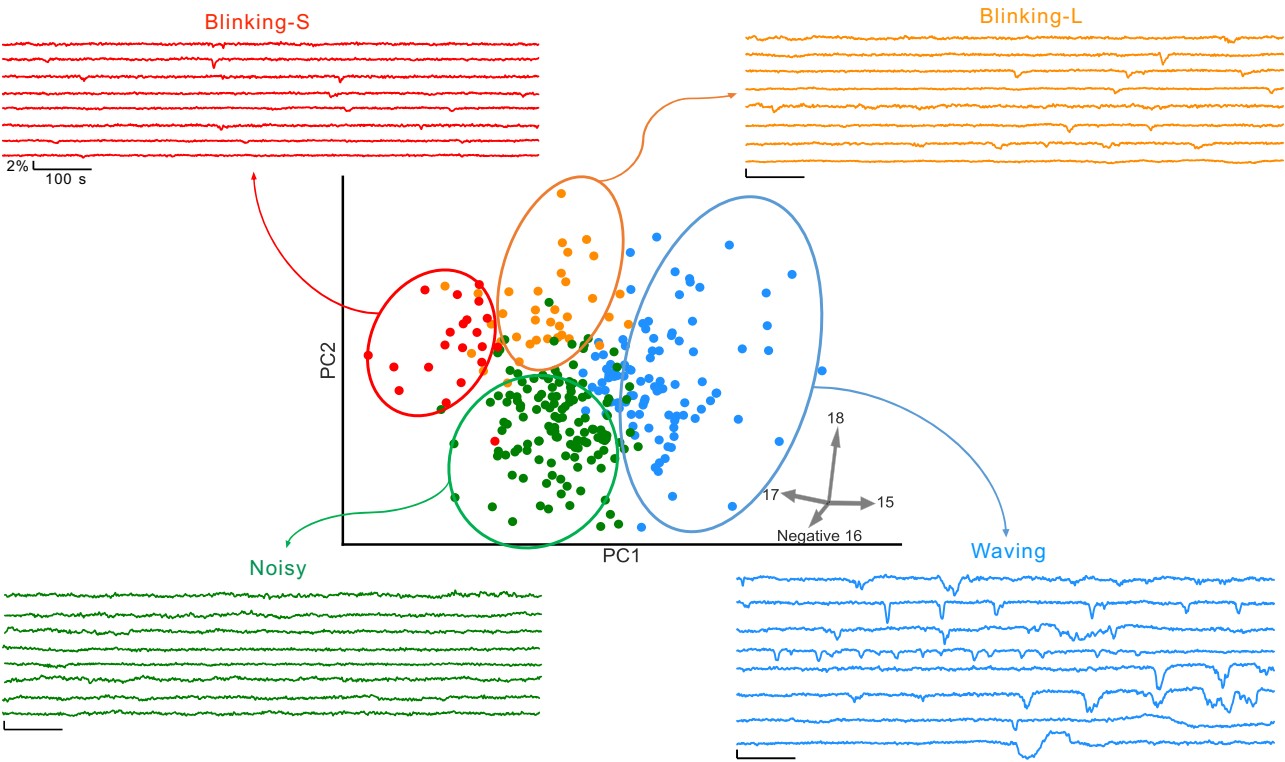

**Fig. 5 Exemplar time series for each DES class.** Representative time series were selected by sorting the values of the feature's pointing in the direction of each type in 2D PC space.

and untreated MCF-10A cells (Fig. 8f). In particular, 7/27 untreated MCF-10A cells and 9/33 TGF-$\beta$-treated MCF-10A cells fell outside of the decision boundary of the best linear separator (Fig. 8f, dashed line). In a feature space combining MDA-MB-231 DESs and MCF-10A DESs, the MCF-10A DESs co-localized over a range of MDA-MB-231 DES classes in 2D PC space, while TGF-$\beta$ treatment increased localization with large-amplitude $V_m$ "waving" MDA-MB-231 cells in the combined PC space (Fig. 8g). In summary, TGF-$\beta$ treatment increased the similarity of MCF-10A $V_m$ dynamics with the large-amplitude "waving" MDA-MB-231 DES class. Thus mesenchymal differentiation correlates with increased $V_m$ fluctuations.

**MDA-MB-231 cells $V_m$ event synchrony.** We observed temporal correlations between events occurring in simultaneously imaged MDA-MB-231 cells. Pairwise comparison of cellular $V_m$ time series revealed a mixture of synchronous and asynchronous events (Fig. 9a), indicating that these temporal correlations were not caused by optical crosstalk. To assess whether the $V_m$ event temporal correlations occurred at rates significantly above chance, we generated event rasters for $V_m$ transients detected in time bins ranging from 1 to 100 s (Fig. 9b, showing 10 s bins). We then calculated the pairwise Pearson correlation coefficient (PCC) for all cells in each FOV. We compared the mean PCC for the cells to the PCC of randomly shuffled rasters, which preserved the cellular event statistics but destroyed any inter-cell temporal correlations. PCC increased overall with bin size as expected, and the PCC for the real event rasters was significantly greater than the PCC of the scrambled rasters for all bins sizes (Fig. 9c). This result indicates a significant inter-cell temporal correlation of MDA-MB-231 $V_m$ events, above that which would be observed by chance for such events randomly distributed in time.

In one instance we observed a wave of transient depolarizations propagating through a subset of cells unidirectionally across the FOV (Fig. 10). The slope of the line fit to the distance as a function of hyperpolarization peak time from the first active cell shows a propagation speed of 27 µm/s (Fig. 10b, Supplemental Movie 2).

## Discussion

The high-throughout, cellular-resolution imaging data indicate that $V_m$ fluctuates dynamically in subpopulations of highly aggressive breast cancer cell lines. Both the event-based and feature-based analyses indicated that the most stereotypical of these events in MDA-MB-231 cells were transient hyperpolarizing "blinks" and "waves." These events featured significant intercellular temporal correlation. Application of the VGSC blocker TTX and two KCa inhibitors, IbTx and apamin, substantially decreased the dynamic $V_m$ activity in MDA-MB-231 cells. The $V_m$ of non-cancerous MCF-10A breast epithelial cells was static compared to that of MDA-MB-231 cells. Treatment of MCF-10A cells with TGF-$\beta$ increased $V_m$ fluctuations and increased the feature-based similarity of their temporal fluctuations to those of MDA-MB-231 cells.

Whole-cell patch clamp is the gold standard for absolute $V_m$ measurement. This technique measured that the steady-state resting $V_m$ of human breast cancer cells is strongly depolarized relative to normal epithelia[15,18–20]. Relatively little, however, had been reported on spontaneous $V_m$ dynamics in human breast cells. The whole-cell current clamp could in principle detect $V_m$ fluctuations, but this has not been reported to date. It is possible that dialysis of the intracellular space by the patch pipette washes out or dampens the molecular machinery underlying the $V_m$ fluctuations. Moreover, the low throughput of patch-clamp recording could hinder the detection of the heterogeneous DESs exhibited by only approximately 1 in 20 cells.

Our detection of $V_m$ fluctuations in subsets of breast cancer and MCF-10A cells was enabled by the application and adaptation of electrochromic voltage dye imaging. Critically, our approach enabled the monitoring of spontaneous $V_m$ fluctuations

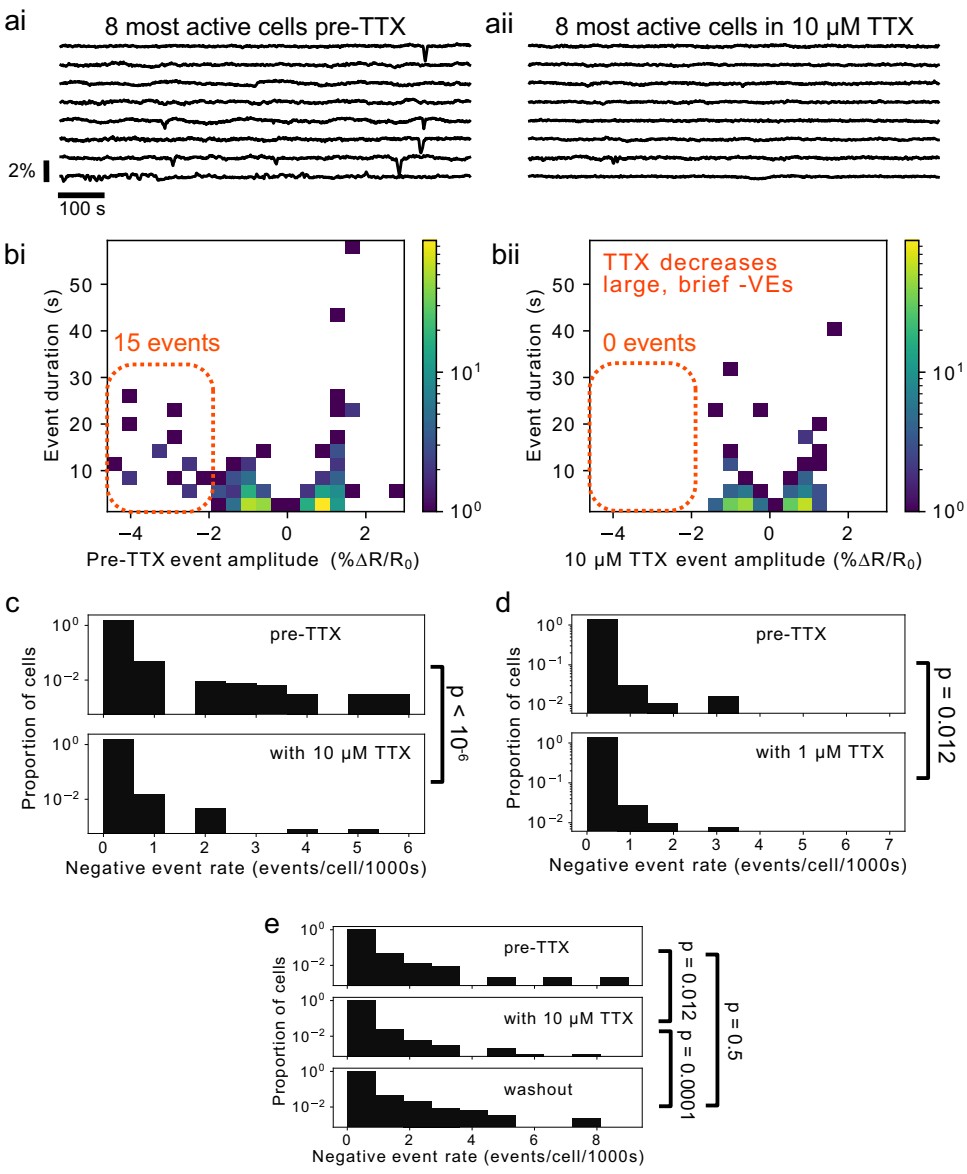

**Fig. 6 TTX decreased $V_m$ fluctuations in MDA-MB-231 cells. ai, ii** Example $V_m$ time courses from the 8 most active cells in acquisitions before (**ai**) and with (**aii**) 10 µM TTX. **bi, bii** Log-scaled 2D histograms of detected events before (**bi**) and with (**bii**) 10 µM TTX. TTX eradicates virtually all large, hyperpolarizing events (−VEs, dashed orange outline). **c** Log-scaled histograms of the −VE event rate per cell for pre- and post-application of 10 µM TTX. The reduction in the mean from $9.5 \times 10^{-5}$ events/cell/s to $1.97 \times 10^{-5}$ events/cell/s (~4× decrease) is significant with $p < 10^{-6}$. **d** The effect of TTX is dose-dependent: 1 µM TTX reduced the mean event rate by a lesser factor of ~2× from $1.04 \times 10^{-4}$ events/cell/s to $4.99 \times 10^{-5}$ events/cell/s ($p = 0.019$). **e** The effect of TTX is reversible. The effects of TTX can be reversed by washing out the toxin, significantly increasing the −VE rate.

in hundreds of cells simultaneously at single-cell resolution. Electrochromic dyes track $V_m$ with sub-microsecond temporal fidelity[35] but are also phototoxic, which limits the exposure duration, rate, and total imaging time (3 ms exposures at 5 Hz over 920 s in our case). In the future, utilization of probes based on other mechanisms such as photo-induced electron transfer[36], or transfection with fluorescent protein-based genetically encoded voltage indicators (GEVIs)[37], may increase the total imaging time and/or rate. In the case of GEVIs, however, the photon budget is often limited by photobleaching, a problem addressed by chemigenetic sensors[38]. Our ratiometric excitation scheme reduced imaging artefacts due to photobleaching, and variations in concentration and volume. It is important to note that voltage dye intensity reports relative changes in $V_m$, not absolute $V_m$. A second limitation of our analysis is that to remove the effects of bleaching and cell movement, we temporally high-pass filter our

$V_m$ traces before analysis via division by the 1000 point rolling average of the time course. This filter precluded the detection of activity varying on timescales slower than approximately 0.01 Hz.

A striking observation is that while all breast cancer lines exhibited $V_m$ fluctuations in normal conditions, MDA-MB-453, MDA-MB-468, and BT-474 were significantly more active as judged by negative event rate frequency (Fig. 2). BT-474 and MDA-MB-453 have been previously characterized as Luminal subtypes, whereas MDA-MB-468 are Basal A[39]. Luminal progenitor cells in the mammary gland are thought to be the cell-of-origin for luminal cancer sub-types[40]. It is tempting to speculate that increased $V_m$ fluctuations may be an inherent property of these progenitors, and/or their transformed derivates. Luminal progenitors can also give rise to basal cancers, and indeed many Basal-A lines in particular are very similar to Luminal lines in that they display epithelial characteristics and have mutational or expression profiles similar to

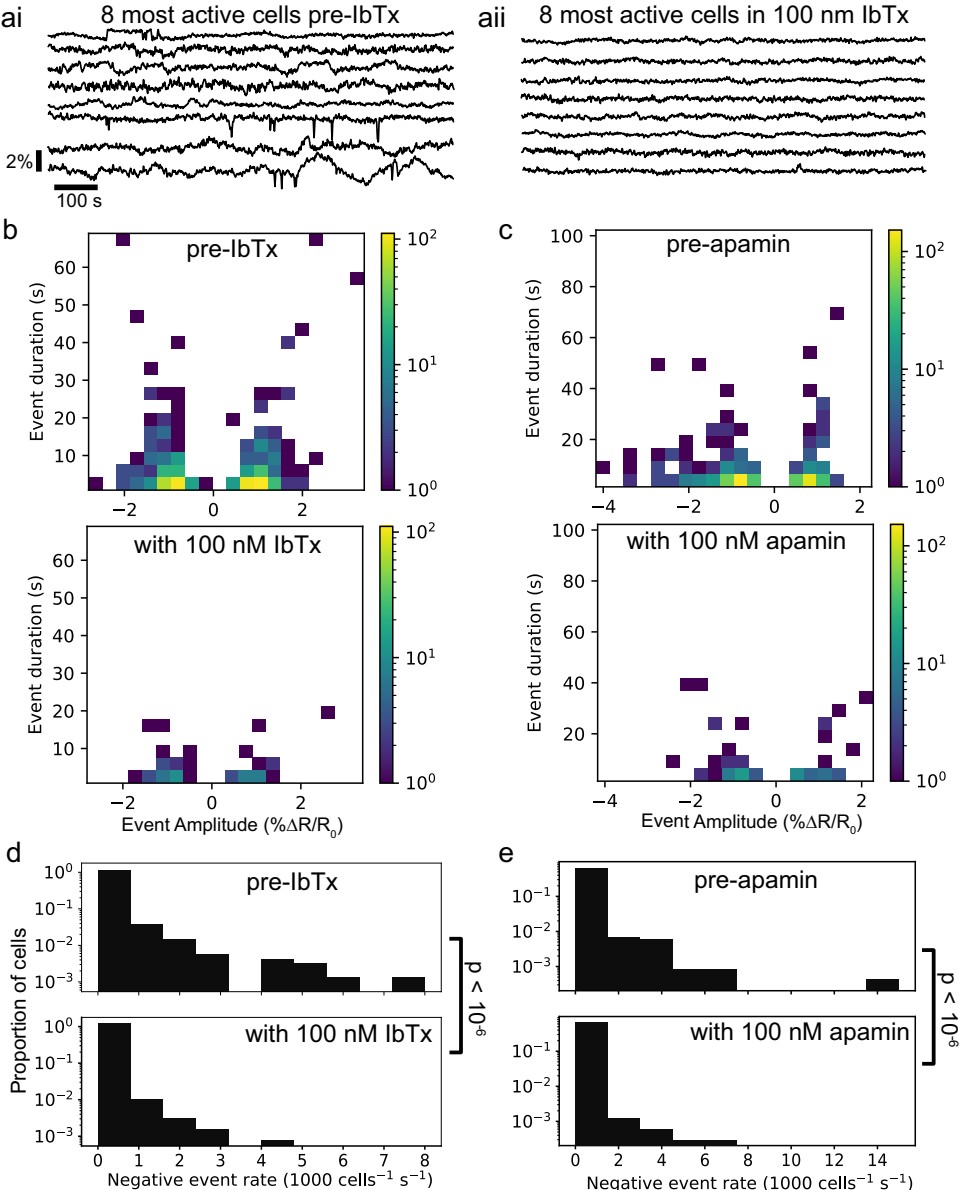

**Fig. 7 KCa channel blockade decreased $V_m$ fluctuations in MDA-MB-231 cells. ai, aii** Example $V_m$ time courses from the 8 most active cells in an acquisition before (**ai**) and with (**aii**) 100 nM IbTx. **b, c** Log-scaled 2D histograms of the amplitude and duration of detected events before (upper) and with (lower) 100 nM IbTx (left) and 100 nM apamin (right). **d, e** Log scaled histograms of the −VE event rate per cell for pre- and post-application of 100 nM IbTx (left) and 100 nM apamin (right).

BRCA1 luminal lines[39,41–43]. Thus the high frequency of $V_m$ fluctuations in MDA-MB-468 may belie that they also originated from a luminal progenitor; unlike other Basal lines (i.e. MDA-MB-231) we examined. Indeed, previous studies have noted morphological similarities between MDA-MB-468 and MDA-MB-453 in that they grow as "grape-like" colonies in 2D and 3D matrices reflecting weak epithelial attachment[44]. In fact, a state of weak attachment, where cells exist in an intermediate between epithelial and mesenchymal forms or a 'hybrid E/M' state[45], may correlate with, or even be conducive to, increased $V_m$ activity. Indeed, we observed induction of EMT by TGF-β of normal MCF-10A epithelial cells led to an increase in $V_m$ fluctuations (Fig. 8). Interestingly, treatment with TGF-β was shown previously also to increase VGSC expression in MCF-10A cells[17]. Potentially cells in this hybrid state retain structures such as gap junctions[46], which could facilitate ion exchange while downregulating cell-cell adhesion (i.e. involving Cadherins).

In MDA-MB-231 cells, a spiking, TTX-sensitive $V_i$ phenotype was recently described through the extracellular multi-electrode array (MEA) recording[28]. In this study, each large area (2 mm²) electrode aggregated signals from 100 s of cells, enabling the detection of rare spiking events from pooled populations. While our 5 Hz imaging rate could not capture the "fast-spiking" (lasting 10 s of milliseconds) activity described by this study, it is possible that the "square-shaped" pulses (lasting up to 7 s) the authors detected arise from the same $V_m$ dynamics reported as blinking/waving in our $V_m$ image series. Our imaging methodology complements high temporal-resolution MEA recordings by enabling high throughput cellular-resolution localization of $V_m$ fluctuations, necessary to detect spatiotemporal patterns, e.g., intercellular $V_m$ wave propagation (Fig. 10). The high throughput of $V_m$ imaging demonstrated here opens a new window onto the 'electro-excitable' pathophysiology of cancer.

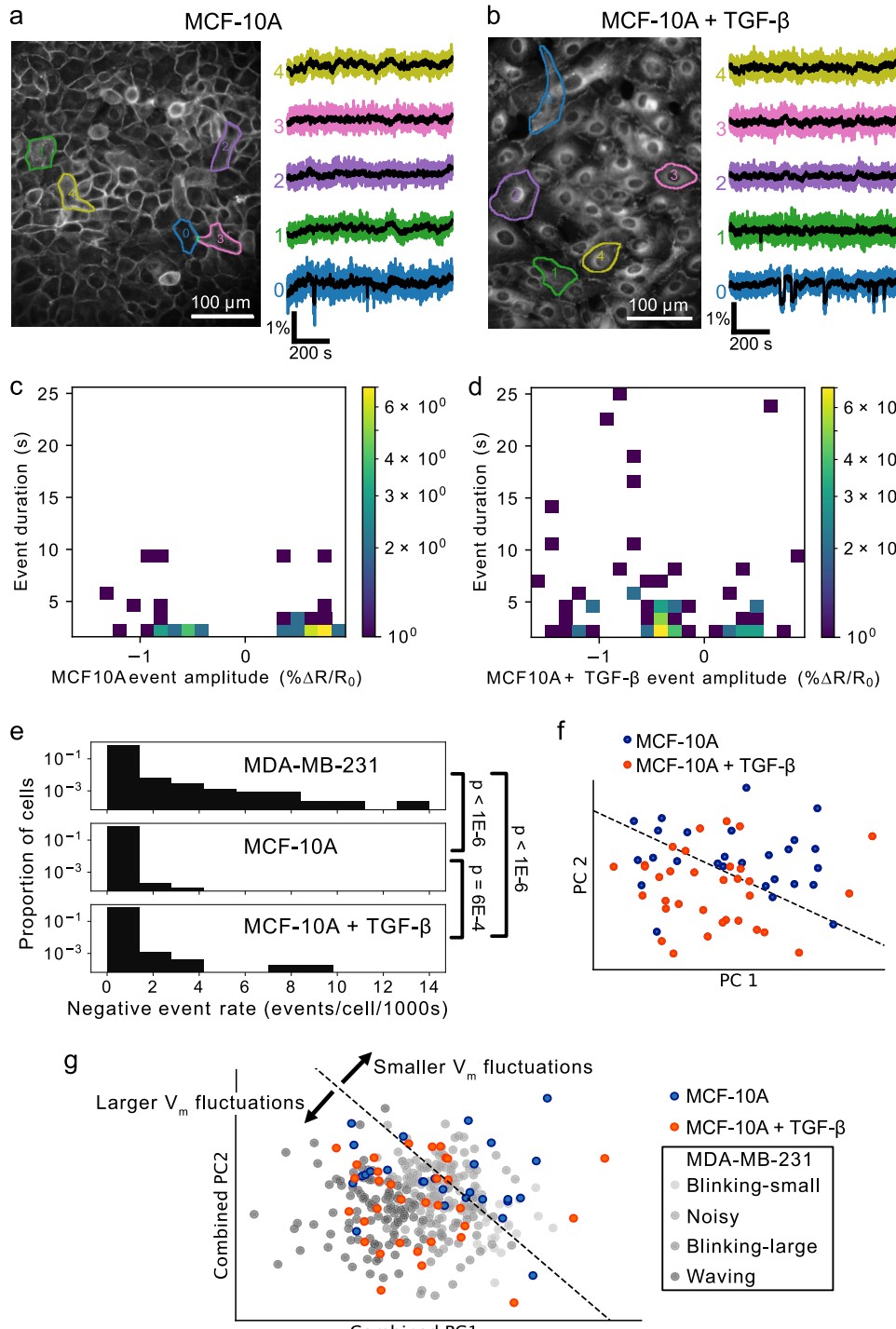

**Fig. 8 Non-cancerous MCF-10A cells exhibit minimal $V_m$ fluctuations which are upregulated by TGF-$\beta$. a, b** Images and cellular $V_m$ time series extracted from MCF-10A cells (**a**) and MCF-10A treated with TGF-$\beta$ (**b**). **c, d** 2D log-scaled amplitude-duration histograms of **c** MCF-10A and **d** MCF-10A+TGF-$\beta$ events. Both MCF-10A and MCF-10A+TGF-$\beta$ exhibit significantly lower event rates than MDA-MB-231 cells. **e** Cell-level comparison of the negative event rate. Incubation of cells with TGF-$\beta$ significantly increases the event rate. **f** Incubation of MCF-10A in TGF-$\beta$ alters the cellular-level DES. The treated and untreated cells locate in different areas of 2D PC space, indicating DES differentiation between these groups. The black dashed line is the best linear separator calculated on the TGF-$\beta$ treated (orange) and untreated (blue) groups. **g** Unlike the untreated MCF-10A which exhibited $V_m$ fluctuations of varying amplitudes, TGF-$\beta$ treatment biased active MCF-10A $V_m$ fluctuations such that they localized near the large-amplitude "waving" MDA-MB-231 class in combined PC space.

Approximately 7% of MDA-MB-231 cells demonstrated transient fluctuations in $V_m$. The large amplitude events were hyperpolarizing 'blinks' and 'waves'. This observation appears in stark contrast to classically excitable tissues like the brain, heart, and muscle that have a resting $V_m$ close to the reversal potential for K$^+$, and which depolarize through the conductance Na$^+$ and/or Ca$^{2+}$ when excited[14]. Resting MDA-MB-231 cells exhibit a relatively depolarized $V_m$, owing at least in part to a Na$^+$ 'window current' conducted by Nav1.5[19,25,47]. The large negative-going events imply that K$^+$ (possibly Cl$^-$) conductance transiently

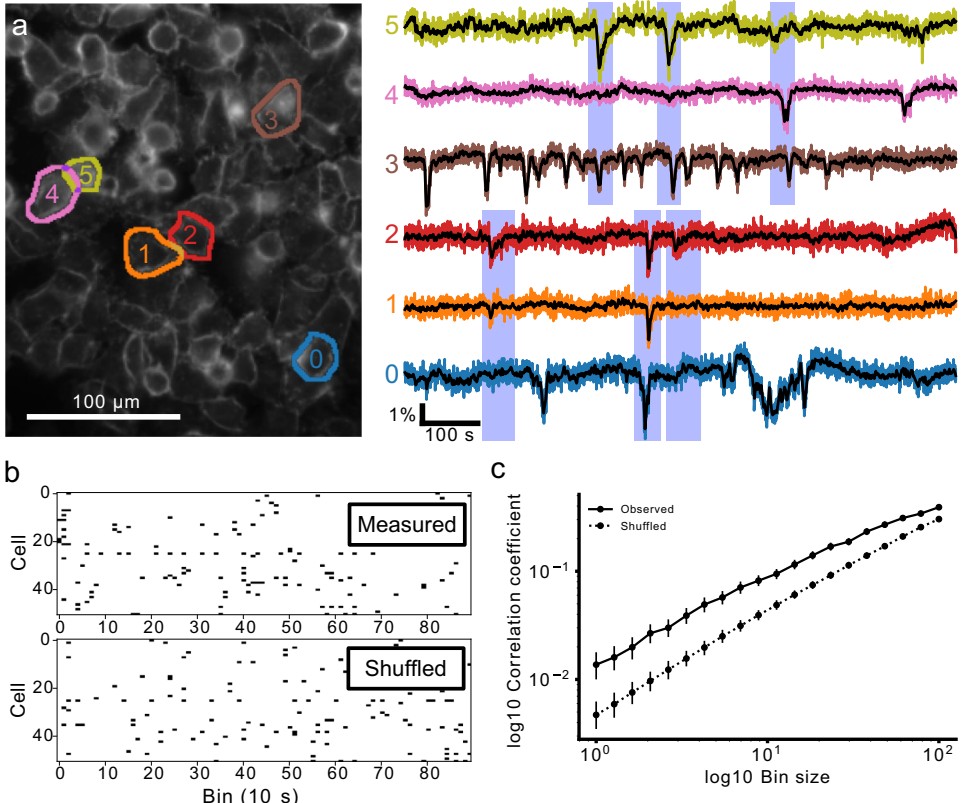

**Fig. 9 Transient activity is temporally correlated in MDA-MB-231 cells. a** Example of temporally correlated cellular time courses from one FOV. Spatially separated cells (0,1,2) and (3,4,5) display synchronized hyperpolarisations. Importantly these synchronizations occur in different subsets of the groups and in spatially separated cells, indicating the synchronization is not simply due to cross-talk in the imaging. **b** Quantification of correlation. Events are put into time bins, generating cellular event rasters. The average pairwise correlation between cells in a recording was calculated. A null hypothesis of zero correlation with the same temporal statistics was generated by temporally shuffling each cell's event raster to obtain an estimate of the expected level of correlation if there were no cellular synchronization. **c** The observed pairwise correlations are significantly higher in the observed data than shuffled data for all bin sizes, indicating that cellular activity is temporally correlated. The points plot the mean, and the bars indicate the 95% confidence interval for the measured and shuffled case for each bin size.

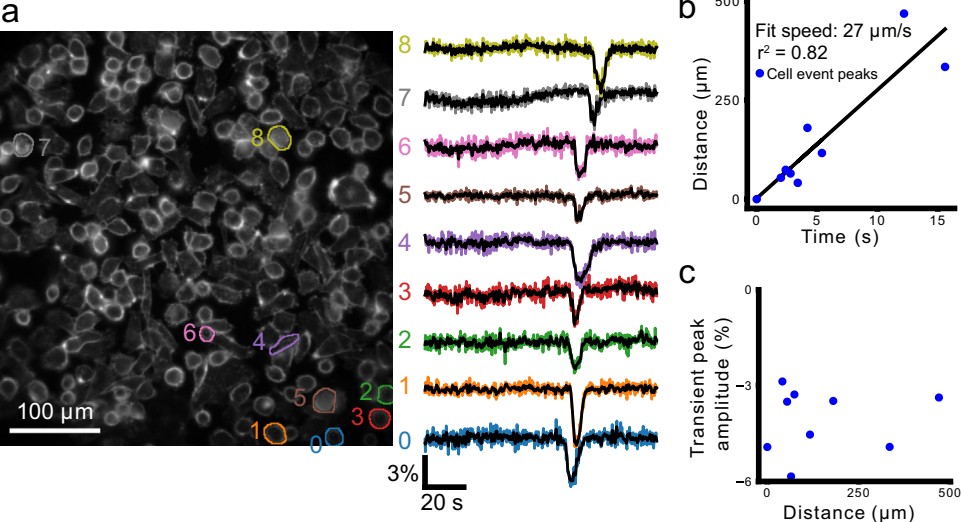

**Fig. 10 A wave of transient hyperpolarizations propagates across the FOV. a** Shows cellular ROIs and their color-corresponding $\Delta R/R_0$ time series (right). **b** Plots distance from the first cell as a function of time at maximum hyperpolarization. The line fit to this relationship shows a propagation speed of 27 μm/s. **c** Plots the peak transient amplitude (%$\Delta R/R_0$) as a function of distance from the cell first showing the transient hyperpolarization.

increases, or that $Na^+$ conductance transiently decreases, hyperpolarizing $V_m$ toward the Nernst potential of $K^+$.

TTX significantly reduced hyperpolarizing transient ($-VE$) frequency in MDA-MB-231 cells. In the presence of TTX, blockade of persistent $Na^+$ currents hyperpolarizes resting $V_m$ in MDA-MB-231 cells[47]. Resting $V_m$ hyperpolarization would decrease the driving force for the conductance of the transient hyperpolarizing current which could account, at least partially, for the reduction of transient hyperpolarizations we detected in the presence of TTX. Further work with, e.g. pharmacological agents and gene silencing, with simultaneous patch-clamp, will be required to fully elucidate the channels and currents mediating the transient hyperpolarizations observed here. TTX-induced hyperpolarization of the resting membrane potential[47] can also decrease the conductance of the voltage- and $Ca^{2+}$-activated BK channels. Indeed, the reduction in transient hyperpolarizations observed with apamin and IbTx implies that the spontaneous hyperpolarizing events can be coupled to $Ca^{2+}$ fluctuations[48] through $Ca^{2+}$-activated $K^+$ channels. Future studies can further elucidate the interplay between $Ca^{2+}$ and $V_m$ through simultaneous calcium and voltage imaging, and by pharmacological or genetic manipulation of KCa conductance during $V_m$ imaging and patch-clamp.

An important distinction between our $V_m$ imaging study and previous patch-clamp recordings is that we report relative changes in $V_m$ rather than absolute $V_m$. Indeed, the subtle few millivolts changes in resting $V_m$ previously detected by patch-clamp (e.g. ref. [47]) are invisible to our imaging methodology, not only due to limited sensitivity but also to the lack of ability to calibrate $V_m$ on an absolute voltage scale based on fluorescence intensity, even intensity ratios. The previous patch-clamp and our fast high-throughput $V_m$ imaging studies, therefore, generate complementary data that will require further work to connect. For instance, one could expect the driving force for transient $K^+$-mediated hyperpolarizations to be greater for individual cells with the most depolarized resting $V_m$. Testing this prediction requires the measurement of absolute $V_m$. In addition to patch-clamp, absolute $V_m$ measurement can be realized optically through $V_m$ fluorescence lifetime imaging (VF-FLIM)[49,50]. However, current VF-FLIM implementations have a temporal resolution < 0.5 Hz and are therefore unable to track the rapid fluctuations characterized here.

The heterogeneity of the $V_m$ fluctuations concurs with the heterogeneous VGSC expression in MDA-MB-231 cells[15]. The largest amplitude fluctuations were transient hyperpolarizing "blinks" and "waves". These hyperpolarizations could directly modulate the amplitude of the "window" current conducted by Nav1.5[19,25], changing intracellular $Na^+$ concentration and downstream $Na^+$-modulated signaling pathways, e.g. SIK[17]. A primary function of the VGSC activity in these cells is controlling proteolysis via pericellular acidification is driven by sodium–hydrogen exchange (NHE1)[51,52]. The fact that the hyperpolarizing $V_m$-driven VGSC activity would occur intermittently in the cells would be suggestive of the following. First, it could ensure better control of the proteolysis, i.e. invasion. Second, it would prevent the excessive influx of $Na^+$ into cells and possible osmotic imbalance that could compromise cell viability[53]. Further work is required to evaluate the potential mechanisms and consequences of $V_m$ fluctuations.

Our results support the notion that $V_m$ and its dynamic behavior are related to cancer cell behavior. First, $V_m$ fluctuations were much more common in cancer cell lines compared with 'normal' breast epithelial MCF-10A cells. Second, blocking VGSC and KCa channel activity dampened the $V_m$ fluctuations in MDA-MB-231 cells. Third, conversely, the dynamic activity of $V_m$ in MCF-10A cells increased after treatment with TGF-$\beta$ is known to induce aggressive behavior in these cells[54]. Although the role of resting $V_m$ in cancer initiation, proliferation, invasion, and metastasis has been characterized in detail (reviewed by ref. [2]), the functional implications of fast, dynamic $V_m$ fluctuations remain to be elucidated. In the future, $V_m$ imaging can be exploited to determine, at a cell-by-cell level, the correlative and causal relationships between $V_m$ behavior and metastatic and structural phenotypes.

## Methods

**Cell culture**. We cultured MDA-MB-231 cells, a kind gift from the laboratory of Dr. Janine Erler, in high glucose DMEM (GIBCO, #41966029) supplemented with 5% FBS (Sigma, #F7524) and penicillin–streptomycin (Sigma, P4333). For imaging, we plated 10k–30k cells on 12 mm collagen-coated glass coverslips (rat tail collagen, Sigma, #122-20). Cells were plated the afternoon prior to imaging. Imaging was performed in phenol red-free Leibovitz's L-15 medium (Thermo Fisher, #21083027), except during the high-$K^+$ control experiments where mammalian physiological saline (MPS) was used instead (described below).

MCF-10A cells were obtained from ATCC. They were cultured in DMEM/F12 (GIBCO, #31331) supplemented with 5% horse serum (GIBCO, #16050), 10 µg/ml insulin (Sigma, #I-1882), 20 ng/ml epidermal growth factor (Sigma, #E-9644), 100 ng/ml cholera toxin (Sigma, #C-8052), 500 ng/ml hydrocortisone (Sigma, #H-0888), and 100 mg/ml penicillin/streptomycin (GIBCO, #15070). Cells were confirmed to be mycoplasma-negative (e-Myco plus Mycoplasma PCR Detection Kit, iNtRON Biotechnology). The passage was carried out using 0.25% trypsin-EDTA (GIBCO) followed by centrifugation (1000 rpm, 4 min) and resuspension in a complete medium. Some sets of MCF-10A cells were cultured in transforming growth factor-$\beta$1 (TGF-$\beta$; Peprotech, #100-21) at 5 ng/mL in complete media for 48–169 h to drive EMT.

The following cell lines were cultured as part of the human breast cancer cell line panel investigated. MDA-MB-468 (a kind gift from the laboratory of Dr. George Poulogiannis, ICR), Cal-51 (a kind gift from the laboratory of Prof. Nicholas Turner, ICR), SUM-159, Hs578t (kind gifts from the laboratory of Dr. Rachel Natrajan, ICR), and MDA-MB-453 (a kind gift from the laboratory of Professor Claire Isacke, ICR) were cultured in high glucose DMEM (GIBCO, #41966-029), supplemented with 10% heat-inactivated FBS (GIBCO, #10500-064) and 1% Penicillin–Streptomycin (GIBCO, #15140-122). T-47D and BT-474, kind gifts from the laboratory of Prof. Nicholas Turner (ICR), were cultured in RPMI 1640 (GIBCO, #11835-063), supplemented with 10% FBS and 1% Penicillin–Streptomycin. Prior to imaging, 12 mm glass coverslips were coated with 50 µg/ml rat tail Type I collagen (Corning, #354236, lot 0295002) in 0.02 M acetic acid. Collagen-coated coverslips were incubated for 2 h at 37 °C. Coverslips were washed twice with PBS and allowed to dry at ambient temperature. All cell lines were passaged using 0.25% trypsin–EDTA (GIBCO), centrifuged (1000 rpm, 5 min), and resuspended in a complete medium. For imaging, 10k–30k cell dilutions were plated on prepared coverslips. Cells were allowed to incubate overnight prior to imaging for optimal attachment. All cell lines were confirmed to be mycoplasma-negative (e-Myco Plus Mycoplasma PCR Detection Kit, iNtRON Biotechnology).

**Imaging**. We prepared a 200 µM stock solution of the electrochromic voltage dye di-4-AN(F)EP(F)PTEA[29] (100 nmol aliquots, Potentiometric Probes) in L-15 solution. The stock solution was kept for a maximum of 3 days after dissolving. Immediately prior to imaging, we gently washed the coverslip-adhered cells three times with warmed L-15 before placing them under the microscope. The coverslip was weighted in place with a tantalum ring and submerged in the dye diluted in L-15 to a final concentration of 3 µM. The cells rested in this configuration for 15 min before imaging. During imaging, cells were maintained at a temperature between 30 and 37 °C by a homemade open-loop water perfusion system or by a closed-loop heated chamber platform (TC-324C, PM-1, Warner Instruments).

Our custom-built widefield epifluorescence microscope formed an image of the cells through a ×25 1.0 NA upright water dipping objective (XLPLN25XSVMP, Olympus) and 180 mm focal length tube lens (TTL180-A) onto a scientific complementary metal-oxide-semiconductor (sCMOS) camera (Orca Flash 4 v2, Hamamatsu). Imaging was performed with two-color sequential excitation and imaged in a single spectral channel. Fluorescence was ratiometrically excited in two channels resulting in opposite-direction voltage signals in the collected emission (Fig. 1c). LEDs were driven using a Cairn OptoLed (P1110/002/000). The first channel was illuminated with 405 nm LED (Cairn P1105/405/LED), filtered with a 405/10 nm bandpass (Semrock LD01-405/10), and combined with a 495 nm long pass dichroic (Semrock FF495-DI03) and an additional 496 nm long pass (Semrock FF01-496/LP). The second channel was illuminated with a 530 nm LED (Cairn P1105/530/LED), filtered with 520/35 nm filter (Semrock FF01-520/35), and combined with a 562 nm long-pass dichroic (Semrock FF562-DI03). Emission was collected through a 650/150 nm bandpass filter (Semrock FF01-650/150) onto the sCMOS camera (Fig. 1a). Images were acquired in Micromanager 2[55] with the Orca Flash 4's 'slow scan' mode, using the global shutter and frame reset with 4 × 4 digital binning. Imaging was performed at 5 Hz. During every image period, a 3-

ms-exposure frame illuminated with each LED was acquired in rapid succession (Fig. S4). Illumination intensities for each channel were approximately matched between each channel and adjusted to give a signal intensity of around 4000 counts/pixel in labeled cell membranes. Intensities were typically between 0.1 and 1.3 mW/mm$^2$ for the blue excitation and 1.5 and 3.4 mW/mm$^2$ for the green excitation.

**Imaging protocols**. We acquired each trial, consisting of sequences of 10,000 frames (5000 dual color-excited acquisitions) at different locations on the coverslip. Imaging locations were selected from confluent areas (median 975 cells/mm$^2$, interquartile range 605, 1247 cells/mm$^2$). We acquired between 1 and 6 trials per coverslip, with each trial occurring in a distinct location. In tetrodotoxin (TTX) experiments, we first imaged 1–2 control trials without TTX ('pre' trials). We then added 1 mM of TTX citrate (Abcam, ab120055) in PBS stock solution to the imaging medium to achieve a final concentration of 1 or 10 μM. 1–4 trials were imaged in the presence of TTX ('post' trials). For certain 10 μM TTX experiments, following 1–2 trials acquired in the presence of TTX, we replaced the TTX-containing medium with regular dye-containing L-15 medium and imaged 1–2 trials in this condition ('washout').

In IbTx and apamin experiments, we first imaged 3 control trials without these agents ('pre-' trials. We then added 24.1 μM of either IbTx citrate (Tocris, #1086) or apamin citrate (Tocris, #1652) in L-15 to the imaging medium to achieve the final concentration of 100 nM. In separate experiments, we treated cells with either 100 nM apamin or IbTx for 45–60 min. The cells were then washed three times with L-15 and voltage dye added as described above to assess $V_m$ dynamics following the washout of the KCa inhibitors.

High potassium wash-in trials were conducted in mammalian physiological saline (MPS[48]), consisting of (in mM): 144 NaCl, 5.4 KCl, 5.6 D-glucose, 5 HEPES, 1 MgCl$_2$, 2.5 CaCl$_2$. Mid-trial, a high-potassium solution was washed in to depolarize the cells for validation of the voltage dye function. This solution was osmotically balanced consisting of (in mM): 49.4 NaCl, 100 KCl, 5.6 D-Glucose, 5 HEPES, 1 MgCl$_2$, 2.5 CaCl$_2$.

**Patch-clamp voltage dye calibration**. We assessed the range of fluorescence change expected for known changes in $V_m$ through whole-cell voltage-clamp and simultaneous voltage imaging. Cells were imaged in phenol red-free Liebovitz's L-15 medium at room temperature. Healthy, dye-labeled cells were selected and patched with pipettes between 3 and 10 MΩ. The pipette contained an Ag/AgCl bathed in intracellular solution (in mM): 130 K-Gluconate, 7 KCl, 4 ATP-Mg, 0.3 GTP-Na, 10 Phosphocreatine-Na, 10 HEPES. Voltage-clamp signals were amplified with a Multiclamp 700B (Molecular Devices) and digitized with Power 1401 (Cambridge Electronic Design) using Spike2 version 9.

Ratiometric imaging was performed as described above but at an increased rate of 100 frames/s. During each imaging trial, $V_m$ was clamped for 1 s epochs at values varying between −60 and +30 mV in 10 mV increments. Fluorescent time courses were extracted from a cellular region of interest (ROI) around the patched cell for both excitation channels. The trials were bleach-corrected and converted to $\Delta F/F_0$ using a linear fit to their time course. We calculated the average blue-to-green excited frame ratio ($\Delta R/R_0$) across trials at each holding potential. A line was fit to $\Delta R/R_0$ vs. $V_m$. The line gradient reflects the sensitivity of $\Delta R/R_0$ to $V_m$ (% change per 100 mV) for each cell ($n = 12$ cells).

**Image processing**. All data analysis was performed in Python 3 using NumPy[56], SciPy[57], Tifffile, Scikit-image[58], Scikit-learn[59], and Pandas[60]. Figures were generated using MatPlotLib[61]. Our dual-color excitation scheme generated image time series interleaving blue and green light-excited frames (Fig. 3a). We subtracted the constant dark value from each frame and separated the time series into two color channels. We applied a pixel-wise high pass filter, rejecting signals slower than 0.01 Hz, to the separated time series. In particular, slowly varying signals (mainly bleaching) were removed from each channel by dividing the stacks pixel-wise by a temporally filtered version of themselves. The filter was a uniform filter of length 1000 points. The filter was symmetric (i.e. time point $t0$ was affected by points $t > t0$ and $t < t0$ equally). These filter-normalized time series were then divided, blue frames by green frames, to find the ratio image for each time point (Fig. 3a). Cells were segmented using CellPose[62], using the default cytoplasmic segmentation model and approximate cellular diameter of 30 pixels. Additional segmentations of active cells that the Cellpose network did not identify were added by hand. The segmented ROIs were eroded with a single (1-round of binary) before extracting the ROI time courses to suppress the effects of movement at the cell edges. For each eroded ROI, we calculated the median of the pixel values at each time point. The mean of the time courses was subtracted and offset so that they were symmetric at about 1.

**Event detection**. We implemented an event detection algorithm to identify significant changes to $\Delta R/R_0$ reflecting the fluctuation of $V_m$. We first calculated the time course of the intra-ROI pixel-wise standard deviation for each eroded ROI. We filtered the median (calculated as above) and standard deviation time courses with a $\sigma = 3$ point (i.e. 0.6 s) Gaussian filter. $V_m$ fluctuation events were identified when the temporally filtered median pixel value diverged from 1, its time average

value, by more than 2.5 times the temporally filtered standard deviation (Fig. 1h). Short events were removed and neighboring events merged by 2 iterations of binary opening and then 2 rounds of binary closing on the detected event array. Where events consisted of positive- and negative-going $\Delta R/R_0$, they were split into entirely positive-going and entirely negative-going events.

**Time series and event inclusion criteria**. Cells were considered dying/dead and excluded from analysis where the raw pixel values increased in brightness by more than 25% during the acquisition in the blue channel, indicating loss of membrane polarization. Events arising from non-voltage-related changes in brightness were identified as simultaneous events and excluded from the analysis. In the MCF-10A image series, where events were very rare, imaging artefacts were identified and excluded where more than three events overlapped by more than 30% in time. In MDA-MB-231 data, events were excluded where more than 5 events overlapped by more than 50% in time. Following event detection, all active cell time series were then evaluated by visual inspection of processed videos to reject events caused by floating dust, focal shifts, or other apparent imaging artefacts. Events satisfying both the automated and manual quality-control measures were analyzed for event frequency, polarity, amplitude and duration.

In the feature-based analysis, we only included trials that completed the full 920 s imaging period (18,279 out of 18,752), and then applied the event detection algorithm described above to identify "active" cells (982 out of 18,279). We then excluded all-time series containing apparent imaging artefacts (dust, etc.). Following quality control and event detection, 297 MDA-MB-231, 28 MCF-10A, and 33 MCF-10A+TGF-$\beta$ time series were admitted to the feature-based analysis pipeline described below (Fig. 4a, b).

**$V_m$ time series clustering analysis**. Complementing the event-based analysis, developed the Cellular DES Pipeline to classify the ROI-extracted time series according to its most salient dynamic features extracted by the Catch-22 algorithm[32]. The analysis realized with the Cellular DES Pipeline provides insight into the time series characteristics beyond simple event detection and quantification, enabling the classification of the heterogeneous $V_m$ dynamics into like clusters.

From the admitted time series, we extracted 22 features from each cellular ROI's median time series with the Catch-22 algorithm (Fig. 4c). After plotting the distribution of the individual features, around 80% (259/324) of cells shared the same value for the feature corresponding to the first minimum of the automutual information function, which we subsequently excluded from the feature list. We rescaled the raw feature values for the remaining 21 features between 0.0001 and 1 and applied the Box-Cox transformation to normalize their distributions. We then rescaled the normalized values to between 0 and 1 to ensure equal weighting into the clustering algorithms. To evaluate the number of dynamic electrical signature (DES) classes, we implemented hierarchical clustering and Gaussian Mixture Modeling (GMM) on the 21 normalized features. Both clustering algorithms generated clusters with silhouette coefficients, which measure subtype dissimilarity[63], decreasing to their lowest levels between 5 and 6 clusters (Fig. 4d). Based on these silhouette scores and on visual inspection of the time series, we chose to sort the time series into four types as this resulted in the most homogeneous classes. Based on the general pattern of each type, we named the DES classes: small blinking, waving, noisy and large blinking (blinking-l).

To select an exemplar time series from each DES class (Fig. 5), we performed Principal components analysis (PCA) and visualized the four classes in a 2-dimensional feature space. Each feature cluster occupies a unique area of the PC space. To identify the exemplar time series of each type, we calculated the components of each feature and drew a vector for each feature's coefficients of PC1 and PC2 (Fig. 5). These vectors, therefore, point to the type exhibiting the corresponding features most saliently. To identify exemplar time series from each type, we sorted the time series according to the feature whose vector points to that type.

**Statistics and reproducibility**. Mean negative event (−VE) rates were compared by bootstrapped significance test, one-sided or two-sided, with the number of cells, the number of cultures ("slips"), and the resulting $p$-values specified in each case. The Supplementary Data contains the data used to generate all figures.

**Reporting summary**. Further information on research design is available in the Nature Research Reporting Summary linked to this article.

## Data availability

The imaging and electrophysiological datasets generated and analyzed during the current study are available from the corresponding author on reasonable request.

## Code availability

The analysis code is available at https://github.com/peq10/cancer_vsd[64].

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

## Acknowledgements

The authors would like to thank Dr. Julia Gallinaro for her advice on the quantification of synchrony within the time courses. This work was supported by a Pump-prime Award from the Integrated Biological Imaging Network (IBIN3AF), the Royal Academy of Engineering under the RAEng Research Fellowships scheme (RF1415/14/26), the Biotechnology and Biology Research Council (BB/R009007/1), a Wellcome Trust Seed Award (Grant No. 201964/Z/16/Z), the Engineering and Physical Sciences Research Council (Grant No. EP/L016737/1 to Imperial College), and by a Cancer Research UK and Stand Up to Cancer UK Program Foundation Award to CB (C37275/1A20146).

## Author contributions

P.Q. devised the image acquisition system and methods with guidance from C.D.A. and A.J.F., P.Q., Y.S., M.B.A.D., C.B., and A.J.F. designed the experiments. P.Q. and Y.S. performed the experiments, analyzed the images. P.Q. designed and P.Q. and Y.S. performed the event-based analysis. M.A.G. and M.B. cultured and plated the MCF-10A, MDA-MB-453, MDA-MB-468, BT-474, CAL-51, Hs-578T, SUM159, and T-47D cells. Y.S. developed the Cellular DES Pipeline with the guidance of C.B., M.B.A.D., A.J.F., and P.Q. Y.S. performed the feature-based analysis with assistance from P.Q. P.Q., Y.S., M.A.G., M.B., C.B., M.B.A.D., and A.J.F. wrote the manuscript. All authors reviewed the manuscript.

## Competing interests

The authors declare the following competing interests: C.D.A. is an owner and employee at Potentiometric Probes LLC, which develops and sells voltage-sensitive dyes. M.B.A.D. is involved in a small biotech company developing ion channel modulators as anti-cancer drugs. The remaining authors declare no competing interests.
