## [Peer Review File · Communications Biology]

Reviewers' comments:

Reviewer #1 (Remarks to the Author):

The authors developed a high-throughput single cell imaging approach to reveal the dynamics of resting membrane potential (V_m) in metastatic (triple-negative) breast cancer cells. The spontaneous hyperpolarization events of TNBC V_m are inhibited by TTX, a selective potent blocker of voltage-gated sodium channel, suggesting the contribution of VGSC to V_m fluctuation. Furthermore, the authors demonstrated that non-cancerous epithelial cells (MCF-10A) lack V_m fluctuation, which can be induced by TGF- β 1, an inducer of the epithelial-to-mesenchymal transition. Based on these data, the authors proposed that V_m fluctuation is acquired during transformation and may participate in carcinogenesis. The research is very interesting. There exist conceptual and experimental issues that need to be addressed.

- Heterogeneous dynamic electrical signatures (DESSs) exhibited in $\sim 7\%$ of MDA-MB-231 cells, as the authors claimed in the Discussion (line 125). Authors used a novel high-throughput imaging approach to detect this low-percent event. Thus, the majority of TNBC cells do not have spontaneous hyperpolarization events. It would help readers to better understand the significance of this finding if authors can discuss the potentially functional role of V_m fluctuations in TNBC initiation, proliferation, invasion, and metastasis.
- VGSC cannot be directly responsible for spontaneous hyperpolarization events, although TTX blocks these events, since VGSC are depolarizing channels. Applying K^+ channel blockers would be mechanistically more meaningful as K^+ channels are hyperpolarizing channels. Have you tried K^+ channel blockers to inhibit hyperpolarization events?
- TGF- β 1 can induce V_m fluctuations in MCF-10A, which is intriguing. The data suggested VGSC activity might be triggered by TGF- β 1 as in cardiac myocytes that TGF- β 1 increased VGSC expression and channel activity. Have you tried TTX to block V_m fluctuations in MCF-10A induced by TGF- β 1?
- Figure 5: TTX on MDA-MB-231. Need to repeat TTX on another TNBC cell line to avoid a potential bias on specific TNBC cell line due to heterogeneity of TNBC cells. What about the effects of TTX on MCF-10A cells (as a control)?
- Movie legends: Movie M1. ...Left, raw "video"..., you mean raw "image"?

Reviewer #2 (Remarks to the Author):

This is an exciting study describing the presence of voltage fluctuations in human breast cancers. While the analysis of these fluctuations is well carried out, additional experiments need to be done to accurately characterize their presence in cancer cells vs. healthy cells.

- Figure 1: It would be helpful to show images of the dye with a Phalloidin cytoskeletal dye to see where it localizes in the cell. Do the cells that display the voltage fluctuations have different morphology?
- It is critical that the authors use additional TNBC cell lines to see if the fluctuations are a unique feature of MDA-MB-231 cells or present in all or a proportion of TNBC cells.
- Induction of EMT: The methods state : Some sets of MCF-10A cells were cultured in Transforming Growth Factor- β 1 (TGF- β ; Peprotech, #100-21) at 5 ng/mL in complete media for 48 - 169 hours to drive EMT. This is a wide range and transition through EMT takes several days, so cells here were likely samples in a range of states. Were all cells analyzed at similar stages of EMT? If not, the authors should analyze on a similar time scale. Further, to support their conclusions that fluctuations are acquired during transformation, additional cell lines would be required.
- The voltage dye images (Fig 6): the signal looks very different for the MCF10A and +TGF β cells – can the authors explain this? Does this impact the readings?
- This sentence in the abstract needs to be toned down: 'These data suggest that the ability to generate V_m fluctuations is acquired during transformation and may participate in oncogenesis'. There is no evidence here that these fluctuations play a functional role and more rigorous analysis needs to be done to show these are actually acquired in transformation.

Reviewer #3 (Remarks to the Author):

In this manuscript, Quicke et al use a high throughput imaging approach to measure the V_m of MDA-MB-231 and MCF-10A cells. They report that a small proportion (~7%) of MDA-MB-231 cells display transient hyperpolarisations, whereas this phenomenon was absent in MCF-10A cells. The results are novel and interesting; however, there are two points that I think the authors could address to strengthen the conclusions.

1. The voltage sensitivity of the dye is very low (5.1%/100 mV) which could be problematic in the context of non-excitabile cells such as these mammary epithelial cell lines. I am not sure the authors can do anything about the inherent limitation with this (and other) V_m dyes. However, I think the results should be interpreted more cautiously in light of this limitation. Other studies have shown relatively modest changes in V_m in MDA-MB-231 cells in response to pharmacological manipulation (e.g. Yang et al 2020). This imaging approach could therefore be missing vital information on subtle V_m changes in the cell population.

2. The effect of TTX, decreasing the frequency of V_m hyperpolarisations, is difficult to reconcile with its effect as a VGSC blocker, which would be expected to reduce Na^+ influx and thus itself hyperpolarise the V_m . This paradox is covered briefly in the discussion, with the hypothesis put forward that TTX would reduce the driving force for K^+ efflux. The problem here is that TTX, at best, hyperpolarises the V_m of this cell line by ~10-15 mV, which from a resting potential of ~-20 mV, is still well above the K^+ equilibrium potential. A more plausible explanation might be that TTX is indirectly inhibiting a hyperpolarising current in this cell line. Indeed, the authors discuss the possible involvement of KCa channels in driving these hyperpolarising fluctuations in the preceding paragraph. A simple experiment to address this mechanism would be to investigate whether KCa channel blockage (e.g. treatment with iberiotoxin etc) inhibits these transients. Such a line of enquiry is actually mentioned in the Discussion. It would significantly strengthen the findings, adding mechanistic insight to this interesting phenomenon.

Reviewer #1 (Remarks to the Author):

The authors developed a high-throughput single cell imaging approach to reveal the dynamics of resting membrane potential (Vm) in metastatic (triple-negative) breast cancer cells. The spontaneous hyperpolarization events of TNBC Vm are inhibited by TTX, a selective potent blocker of voltage-gated sodium channel, suggesting the contribution of VGSC to Vm fluctuation. Furthermore, the authors demonstrated that non-cancerous epithelial cells (MCF-10A) lack Vm fluctuation, which can be induced by TGF- β 1, an inducer of the epithelial-to-mesenchymal transition. Based on these data, the authors proposed that Vm fluctuation is acquired during transformation and may participate in carcinogenesis. The research is very interesting. There exist conceptual and experimental issues that need to be addressed.

We thank Reviewer #1 for their enthusiasm and clear assessment of our manuscript, and for the constructive recommendations to improve it. We have implemented and responded to the recommendations and detail these changes below.

- *Heterogeneous dynamic electrical signatures (DESSs) exhibited in ~7% of MDA-MB-231 cells, as the authors claimed in the Discussion (line 125). Authors used a novel high-throughput imaging approach to detect this low-percent event. Thus, the majority of TNBC cells do not have spontaneous hyperpolarization events. It would help readers to better understand the significance of this finding if authors can discuss the potentially functional role of Vm fluctuations in TNBC initiation, proliferation, invasion, and metastasis.*

Although the role of resting Vm in cancer initiation, proliferation, invasion, and metastasis has been characterized in detail [1], the functional implications of *fast, dynamic* Vm fluctuations remains to be elucidated. Our new data from several breast cancer cell lines, all displaying more dynamic Vm than the non-tumorigenic MCF-10A cell line, is consistent with Vm dynamics playing a functional role in cancer progression. The two types of channels shown to be involved in the transients (voltage-gated sodium channels re the effect of TTX and calcium-activated potassium channels re Iberiotoxin and apamin) have already been shown to contribute to several types of metastatic cell behavior (reviewed in [2]). Future work will leverage our optical Vm monitoring methodology in conjunction with assays for proliferation, migration, and invasion, transcript/proteomic data sets, and optogenetics to uncover the specific correlative and causal links between Vm dynamics and the different components of cancer progression. These points are now included in the Discussion lines 246 - 253.

- *VGSC cannot be directly responsible for spontaneous hyperpolarization events, although TTX blocks these events, since VGSC are depolarizing channels. Applying K⁺ channel blockers would be mechanistically more meaningful as K⁺ channels are hyperpolarizing channels. Have you tried K⁺ channel blockers to inhibit hyperpolarization events?*

We agree that potassium is the most likely ion species to drive transient Vm hyperpolarization. In view of previous work showing that MDA-MB-231 cells exhibit spontaneous calcium oscillations [3], we suspected

that calcium-gated K⁺ (KCa) channels may transduce the calcium fluctuations into the transient hyperpolarizations we have observed. We have therefore characterized the V_m dynamic sensitivity to blockade of BK-KCa channels (100 nM Iberiotoxin) and SK-KCa channels (100 nM Apamin). Each of these caused a 5-fold decrease in negative event rate. These effects were highly significant ($p < 10^{-6}$) and washed out. We have added these new findings to the Results lines 115 – 128 (including new Figure 7), the Discussion lines 221-226, and the Methods lines 315-319.

- *TGF-β1 can induce V_m fluctuations in MCF-10A, which is intriguing. The data suggested VGSC activity might be triggered by TGF-β1 as in cardiac myocytes that TGF-β1 increased VGSC expression and channel activity. Have you tried TTX to block V_m fluctuations in MCF-10A induced by TGF-β1?*

We have not assessed the effects of TTX on TGF-β1-treated MCF-10A cells. Multi-electrode recordings previously showed that TTX blocks “spiking” activity in MDA-MB-231 cells [4]. TTX’s effect on our optically detected V_m transients importantly corroborates these data. The low frequency of hyperpolarizing events in the TGF-β1-treated MCF-10A cells (Figure 8E) means that many more TTX-trials would be required than for MDA-MB-231 cells to obtain the statistical power necessary to assess TTX’s effect. At this point the cost and time required for this are prohibitive. Therefore, we limit this initial pharmacology to the electrically active MDA-MB-231 line.

- *Figure 5: TTX on MDA-MB-231. Need to repeat TTX on another TNBC cell line to avoid a potential bias on specific TNBC cell line due to heterogeneity of TNBC cells. What about the effects of TTX on MCF-10A cells (as a control)?*

Due to budget and time limitations, we have not performed pharmacological characterization of the MCF-10A cells or the other newly included breast cancer cell lines. Our results indicate that hyperpolarizing events characterize the membrane potential dynamics of several TNBC as well as luminal and basal-A cell lines. These cell lines exhibit substantial known diversities in morphology and ion channel expression. Their full characterization and comparison will be important however outside the scope of our current study. Here we describe a novel, high-throughput optical strategy for studying breast cancer V_m through the characterization of the MDA-MB-231 cell line.

- *Movie legends: Movie M1. ...Left, raw “video”..., you mean raw “image”?*

Yes, that should read raw “image”, and is fixed in the revision. Thank you.

Reviewer #2 (Remarks to the Author):

This is an exciting study describing the presence of voltage fluctuations in human breast cancers.

We thank Reviewer 2 for their careful reading of our manuscript and points to improve, which we address here below.

While the analysis of these fluctuations is well carried out, additional experiments need to be done to accurately characterize their presence in cancer cells vs. healthy cells.

- *Figure 1: It would be helpful to show images of the dye with a Phalloidin cytoskeletal dye to see where it localizes in the cell.*

The extracellularly-loaded di-4-AN(F)EP(F)PTEA voltage dye localizes to the outer plasma membrane as now clarified in the Results lines 53-55. This localization is necessary for sensing the variations in transmembrane potential. Fixing the cells to image Phalloidin would permeabilize the cell and severely disrupt the voltage dye fluorescence.

Do the cells that display the voltage fluctuations have different morphology?

This is an excellent question and a work in progress that we are pursuing with much interest although beyond the scope of this manuscript.

- *It is critical that the authors use additional TNBC cell lines to see if the fluctuations are a unique feature of MDA-MB-231 cells or present in all or a proportion of TNBC cells.*

Whether the V_m fluctuations are a general feature of TNBC or indeed other aggressive breast cancer types is an important question. Therefore, we have imaged and include in the revised manuscript imaging results for lines MDA-MB-453, MDA-MB-468, BT-474, CAL-51, Hs-578T, SUM159, and T-47D. Like MDA-MB-231, each of these lines exhibited significantly higher negative event rates than the non-tumorigenic MCF-10A line ($p \leq 0.01$). We have added these new results to lines 69-80, to the new Figure 2, to the Discussion lines 185-200, and to the Methods lines 270-282 of the revised manuscript. In this revision we carefully avoid implying that these fluctuations are a special feature of TNBCs, as the fluctuations were also observed in luminal, basal A and basal B cell lines.

- *Induction of EMT: The methods state : Some sets of MCF-10A cells were cultured in Transforming Growth Factor- β 1 (TGF- β ; Peprotech, #100-21) at 5 ng/mL in complete media for 48 - 169 hours to drive EMT. This is a wide range and transition through EMT takes several days, so cells here were likely samples in a range of states. Were all cells analyzed at similar stages of EMT? If not, the authors should analyze on a similar time scale. Further, to support their conclusions that fluctuations are acquired during transformation, additional cell lines would be required.*

Comparing negative Vm event rates between earlier and later TGF-beta treated time points revealed no statistically significant difference. As hyperpolarization events in MCF-10A cells occurred rarely in any case, our comparison between timepoints does not have the statistical power required to resolve a difference if present.

Restricting the comparison in untreated MCF-10A cells and those treated within a narrower time window of 100-169 hours still shows a statistically significant ($p < 0.001$) decrease in negative event rate with TGF-beta treatment. We are therefore confident in concluding that TGF-beta treatment leads to the generation of Vm fluctuations and do not make any further claims at present. Future work can interestingly utilize our imaging methodology to characterize the evolution of DES, and concomitant changes in morphology, as a function of EMT stage.

- *The voltage dye images (Fig 6): the signal looks very different for the MCF10A and +TGFbeta cells – can the authors explain this? Does this impact the readings?*

TGF-beta has known effects on cellular morphology [5]. Apart from this, the labeling of the extracellularly-applied voltage dye is expected to behave the same in the various cell lines by localizing to the outer plasma membrane. The imaging is ratiometric therefore additionally mitigating differences in labeling efficiency (which in any case are not expected).

- *This sentence in the abstract needs to be toned down: ‘These data suggest that the ability to generate Vm fluctuations is acquired during transformation and may participate in oncogenesis’. There is no evidence here that these fluctuations play a functional role and more rigorous analysis needs to be done to show these are actually acquired in transformation.*

We agree that this was a bold statement considering the analysis of a single breast cancer cell line. In the new version, in light of the new measurements of Vm activity across several cell lines, we have changed this sentence to: “These data suggest that the ability to generate Vm fluctuations may be a property of hybrid epithelial-mesenchymal cells or those originated from luminal progenitors.”

Reviewer #3 (Remarks to the Author):

In this manuscript, Quicke et al use a high throughput imaging approach to measure the Vm of MDA-MB-231 and MCF-10A cells. They report that a small proportion (~7%) of MDA-MB-231 cells display transient hyperpolarisations, whereas this phenomenon was absent in MCF-10A cells. The results are novel and interesting; however, there are two points that I think the authors could address to strengthen the conclusions.

We thank Reviewer 3 for their interest and careful reading of our manuscript and comments to improve it. We address each below.

1. *The voltage sensitivity of the dye is very low (5.1%/100 mV) which could be problematic in the context of non-excitabile cells such as these mammary epithelial cell lines. I am not sure the authors can do anything about the inherent limitation with this (and other) Vm dyes. However, I think the results should be interpreted more cautiously in light of this limitation. Other studies have shown relatively modest changes in Vm in MDA-MB-231 cells in response to pharmacological manipulation (e.g. Yang et al 2020). This imaging approach could therefore be missing vital information on subtle Vm changes in the cell population.*

Voltage dyes have limited sensitivity (due to lower signal-to-noise ratio) and sampling rate compared to whole-cell patch clamp, and generally lack the ability to report absolute membrane potential, which we consider carefully in interpreting the results. An important distinction between our voltage imaging work and previous patch-clamp recordings is that our study reports spontaneous Vm fluctuations rather than absolute resting Vm or currents in response to step changes in command voltage. Indeed, the subtle changes in resting membrane potential previously detected by patch-clamp are invisible to our imaging methodology, not only due to limited sensitivity, but also to the lack of ability to calibrate Vm on an absolute voltage scale based on fluorescence intensity, even intensity ratios. The previous patch-clamp studies and our fast voltage dye experiments therefore generate complementary data that will require further work to understand together. For instance, one could expect the driving force for a transient K⁺-mediated hyperpolarization to be greater for individual cells with the most depolarized resting Vm. While fast, intensity-based measurements alone cannot test this prediction, its future combination with patch clamp or calibrated fluorescence lifetime measurements will. We now discuss this in the Discussion lines 227-236.

2. *The effect of TTX, decreasing the frequency of Vm hyperpolarisations, is difficult to reconcile with its effect as a VGSC blocker, which would be expected to reduce Na⁺ influx and thus itself hyperpolarise the Vm. This paradox is covered briefly in the discussion, with the hypothesis put forward that TTX would reduce the driving force for K⁺ efflux. The problem here is that TTX, at best, hyperpolarises the Vm of this cell line by ~10-15 mV, which from a resting potential of ~-20 mV, is still well above the K⁺ equilibrium potential.*

Due to the limited sensitivity of the voltage imaging to V_m , even a 10 mV reduction in driving force would be expected to reduce the frequency of *detected* hyperpolarization events, which is why we hypothesize that reduced driving force contributes at least partially to the observed reduction in event rate.

A more plausible explanation might be that TTX is indirectly inhibiting a hyperpolarising current in this cell line. Indeed, the authors discuss the possible involvement of KCa channels in driving these hyperpolarising fluctuations in the preceding paragraph. A simple experiment to address this mechanism would be to investigate whether KCa channel blockage (e.g. treatment with iberiotoxin etc) inhibits these transients. Such a line of enquiry is actually mentioned in the Discussion. It would significantly strengthen the findings, adding mechanistic insight to this interesting phenomenon.

We have now characterized the V_m dynamic sensitivity to blockade of voltage-sensitive BK-KCa channels (100 nM Iberiotoxin) and voltage-insensitive SK-KCa channels (100 nM Apamin). Each of these caused a 5-fold decrease in negative event rate. These reductions are highly significant ($p < 1 \times 10^{-6}$) and washed out. These results indicate that KCa channels indeed play a role in generation of the transient hyperpolarizations, likely link to spontaneous calcium oscillations [3]. The drastic reduction of hyperpolarizing events with BK-channel blocker IbTx implies an additional potential action of TTX which is known to hyperpolarize membrane potential in MDA-MB-231 cells. As BK channels are both calcium and voltage-sensitive, a TTX-induced hyperpolarization of resting membrane potential would also reduce BK channel conductance. We have added these new findings to the Results lines 115 – 128 (including new Figure 7), the Discussion lines 221-226, and the Methods lines 316-320B.

References

- [1] Yang, M., & Brackenbury, W. J. (2013). Membrane potential and cancer progression. *Frontiers in Physiology*, 4. doi:10.3389/fphys.2013.00185
- [2] Payne, S. L., Levin, M., & Oudin, M. J. (2019). Bioelectric Control of Metastasis in Solid Tumors. *Bioelectricity*, 1(3), 114–130. doi:10.1089/bioe.2019.0013
- [3] Rizaner, N., Onkal, R., Fraser, S. P., Pristerá, A., Okuse, K., & Djamgoz, M. B. A. (2016). Intracellular calcium oscillations in strongly metastatic human breast and prostate cancer cells: control by voltage-gated sodium channel activity. *European Biophysics Journal*, 45(7), 735–748. doi:10.1007/s00249-016-1170-x
- [4] Ribeiro, M., Elghajiji, A., Fraser, S. P., Burke, Z. D., Tosh, D., Djamgoz, M. B. A., & Rocha, P. R. F. (2020). Human Breast Cancer Cells Demonstrate Electrical Excitability. *Frontiers in Neuroscience*, 14. doi:10.3389/fnins.2020.00404
- [5] Milano, D. F., Natividad, R. J., Saito, Y., Luo, C. Y., Muthuswamy, S. K., & Asthagiri, A. R. (2016). Positive Quantitative Relationship between EMT and Contact-Initiated Sliding on Fiber-like Tracks. *Biophysical Journal*, 111(7), 1569–1574. doi:10.1016/j.bpj.2016.08.037

REVIEWERS' COMMENTS:

Reviewer #1 (Remarks to the Author):

In the revised manuscript, the authors performed additional experiments and added two new figures to provide stronger supporting evidence for the main conclusions, which addressed my comments and suggestions.

No further comments.